# Myricetin Exerts Anti-Obesity Effects through Upregulation of SIRT3 in Adipose Tissue

**DOI:** 10.3390/nu10121962

**Published:** 2018-12-12

**Authors:** Seun Akindehin, Young-Suk Jung, Sang-Nam Kim, Yeon-Ho Son, Icksoo Lee, Je Kyung Seong, Hyun Woo Jeong, Yun-Hee Lee

**Affiliations:** 1College of Pharmacy, Yonsei University, Incheon 21983, Korea; akindehin@gmail.com (S.A.); syh8116@naver.com (Y.-H.S.); 2College of Pharmacy, Pusan National University, Busan 46241, Korea; youngjung@pusan.ac.kr; 3College of Pharmacy and Research Institute of Pharmaceutical Sciences Seoul National University, Seoul 08826, Korea; sangnamik@nate.com; 4College of Medicine, Dankook University, Cheonan-si, Chungcheongnam-do 31116, Korea; icksoolee@dankook.ac.kr; 5Laboratory of Developmental Biology and Genomics, BK21 Program Plus for Advanced Veterinary Science, Research Institute for Veterinary Science, College of Veterinary Medicine, Seoul National University, Seoul 08826, Korea; snumouse@snu.ac.kr; 6Korea Mouse Phenotyping Center (KMPC), Seoul 08826, Korea; 7Vital Beautie Division, Amorepacific R&D Center, 314-1 Bora-dong, Giheung-gu, Yongin-si, Gyeonggi-do 17074, Korea; misterjay@amorepacific.com

**Keywords:** myricetin, adipose tissue, SIRT3, anti-obesity, mitochondria

## Abstract

Myricetin is a biologically active natural polyphenol with beneficial effects on metabolic health. This study aimed to examine the effects of myricetin on the expression levels of genes involved in lipolysis and mitochondrial respiration in adipocytes and the anti-obesity potential of myricetin. The results indicated that myricetin reduced triglyceride (TG) content and increased mitochondrial content and oxygen consumption rate (OCR) in adipocytes in vitro. To determine anti-obesity effect of myricetin, C57BL6/J mice were fed a high-fat diet (HFD) for eight weeks and then treated with myricetin (10 mg/kg) for 2 weeks. The in vivo treatment of myricetin reduced body weight by 11%. Furthermore, it improved the glucose tolerance, and increased fatty acid consumption of HFD-fed mice. Myricetin treatment increased Sirt3 expression and reduced the acetylation of mitochondrial proteins in adipose tissue. Finally, the knockdown of Sirt3 in adipocytes reduced the myricetin-induced increase in mitochondrial oxygen consumption rate by about 27% compared to controls. Our results indicated that myricetin exerted anti-obesity effects through the upregulation of Sirt3 expression and mitochondrial metabolism in adipose tissue.

## 1. Introduction

We are currently in the midst of a worldwide epidemic of obesity and its related metabolic diseases [1]. Obesity is defined as excess body fat mass, whereas metabolic syndrome has been described as an adipose tissue disease with impaired energy homeostasis and chronic inflammation [2]. Thus, it is important to understand the mechanisms that promote and maintain metabolically healthy adipose tissue.

Adipose tissue is a specialized organ that can store excess energy as triglycerides (TGs) and mobilize TGs into free fatty acids during negative energy balance, such as fasting and calorie restriction [3]. It is well established that the mobilization of free fatty acids is primarily regulated by the sympathetic activation of protein kinase A (PKA)-dependent cytosolic lipases [3].

In general, adipose tissue can be categorized into anabolic white adipose tissue and catabolic brown adipose tissue [3]. In contrast to white adipose tissue, brown adipose tissue contains a high level of mitochondria and uniquely expresses uncoupling protein 1 (UCP1) for heat production [3]. Importantly, the mitochondrial content of white adipose tissue can be increased by thermogenic stimuli [2]. Thus, the upregulation of mitochondrial energy metabolism in adipose tissue has been considered as a therapeutic strategy to increasing energy expenditure [4].

Myricetin is a naturally occurring polyphenol in diverse plants, fruits, and teas [5], offering potential metabolic health benefits for the treatment of insulin resistance and type 2 diabetes in rodents model [6,7,8]. The biological effects of myricetin are reported to be mediated by several cellular signaling pathways that can mimic the effects of calorie restriction, such as the mammalian target of rapamycin (mTOR) pathway [9], 5’adenosin monophosphate-activated protein kinase (AMPK) signaling [10], and Sirtuin regulation [11]. Thus, the anti-obesity and anti-diabetic potential of myricetin underlie its beneficial effects on metabolism [12]; however, the role of myricetin in adipocyte lipid metabolism has not been fully investigated.

Sirtuins are nicotinamide adenine dinucleotide (NAD+) dependent enzymes with deacetylating, desuccinylating, and demalonylating activities, and an ability to increase mitochondriogenesis and mitochondrial respiration [13]. Seven sirtuin genes have been identified in mammals, [13], with distinct subcellular locations conferring a different enzymatic ability and biological influence on each encoded protein [13]. SIRT1 was shown to de-acetylate PGC1-alpha, and increase mitochondrial biogenesis and metabolism, whereas mitochondrial Sirt3 participates in the control of fatty acid metabolism [14]. Thus, it has been suggested that the regulation of the sirtuin genes may have a role in lipid metabolism of adipose tissue [15]; however, the role of myricetin on sirtuin activity in adipose tissue remains unknown.

Here, we investigated the effect of myricetin on the TG content, expression levels of genes involved in lipolysis, and mitochondrial respiration. We employed a high-fat diet (HFD)-fed mouse model to determine the anti-obesity effects of myricetin. The current study demonstrated that myricetin increased mitochondrial oxidative metabolism in adipose tissue through the upregulation of Sirt3 expression.

## 2. Materials and Methods

### 2.1. Animals

All animal experiments were conducted at Yonsei University and were carried out in in strict compliance with the guidelines for humane care and use of laboratory animals specified by the Ministry of Food and Drug Safety. All animal protocols were approved by the Institutional Animal Care and Use Committees at Yonsei University (IACUC-201803-704-01). Mice were housed at 22 °C and maintained on a 12-h light/12-h dark cycle with free access to food and water. Six C57BL mice (6-weeks old, male) were purchased from Central Lab Animal Inc (Seoul, Korea). For the HFD experiments, 60% fat diet [16,17] (Research Diets # D12492, protein: 20% kcal, fat: 60% kcal, carbohydrate: 20% kcal, energy density: 5.21 kcal/g) was introduced at 6 weeks of age and continued for 8 weeks. Mice fed with a high fat diet (HFD) for 8 weeks were treated with myricetin dissolved in saline by intraperitoneal injection (Sigma, St. Louis, MO, USA, 10 mg/kg body weight) once a day for a period of 2 weeks.

For glucose tolerance test, mice were given D-glucose (2 g/kg body mass, sigma) by intraperitoneal injection, and glucose concentrations were measured at indicated time points. Metabolic measurement was obtained using indirect calorimetry system (PhenoMaster, TSE system, Bad Homburg, Germany), as described previously [18]. Serum TGs and free fatty acid (FFA) were measured by using commercially available TG colorimetric assay kit (Cayman Chemical, Ann Arbor, MI, USA) and FFA quantification colorimetric kit (WAKO), according to the manufacturer’s instructions.

Subcellular fractionation was performed as described previously [19]. Briefly, from adipose tissue homogenates in fractionation buffer (containing 3 mM HEPES (pH7.4), 210 mM mannitol, 70 mM sucrose and 0.2 mM EDTA), cells and debris pellets were removed after centrifugation at 500× *g* for 10 min. The Mitochondria Isolation Kit for Tissue (Thermo Scientific, Waltham, MA, USA) was used to obtain mitochondrial fraction.

### 2.2. Cell Cultures

The C3H10T1/2 cells (ATCC (Manassas, VA, USA)) were cultured, as previously described [20]. Briefly, the cells were cultured in growth medium (Dulbecco’s modified Eagle’s medium (DMEM: Sigma) supplemented with 10% fetal bovine serum (FBS, Gibco Thermo Fisher Scientific, Waltham, MA, USA) and 1% penicillin/streptomycin (Thermo Fisher, Waltham, MA, USA), and then exposed to adipogenic differentiation medium (DMEM supplemented with 10% FBS, BMP4 (20 ng/mL, R&D system, Minneapolis, MN, USA), indomethacin (0.125 mM, Cayman, Ann Arbor, MI, USA), isobutylmethylxanthine (2.5 mM, IBMX, Cayman), dexamethasone (1 μM, Cayman, Ann Arbor, MI, USA), insulin (10 μg/mL, Sigma, St. Louis, MO, USA) and triiodothyronine (T3, 1 nM, Cayman, Ann Arbor, MI, USA) for 3 days. For the maintenance of adipogenic differentiation, the cells were exposed to DMEM containing 10% FBS, 10 μg/mL insulin (Sigma, St. Louis, MO, USA) and 1 nM triiodothyronine (T3, Cayman, Ann Arbor, MI, USA) for 3 days.

Fully differentiated adipocytes were exposed to DMEM supplemented with 10% FBS overnight and then treated with indicated concentration of myricetin (purity > 95%; Sigma). The cells were cultured in Earle’s balanced Salt Solution (EBSS, Thermo Fisher, Waltham, MA, USA) for nutrient starvation. The cells were treated with 1mM 8-bromoadenosine 3’5’-cyclic monophosphate (8-Br-cAMP) for PKA activation. Intracellular TG content was determined using a commercially available TG colorimetric assay kit (Cayman, Ann Arbor, MI, USA) according to the manufacturer’s instructions. Mitochondrion-labeling in live cells was performed using red-fluorescent mitochondrion-selective probe MitoTracker Red CMXRos (Thermo Fisher, Waltham, MA, USA). For Sirt3 knockdown, siRNA targeting Sirt3 (Sigma, St. Louis, MO, USA) was transfected into adipocytes differentiated from C3H10T1/2 cells, using Lipofectamin2000 (Thermo Fisher, Waltham, MA, USA).

### 2.3. Gene Expression Analysis

Quantitative real time polymerase chain reaction (qRT-PCR) was performed, as previously described [20]. Briefly, RNA was extracted using the TRIzol^®^ reagent (Invitrogen, Carlsbad, CA, USA), and RNA was reverse transcribed using a cDNA synthesis kit (High-capacity cDNA Reverse Transcription kit; Applied Biosystems, Foster City, CA, USA). cDNA was subjected to qPCR by using iQ SYBR Green Supermix (Bio-Rad, Hercules, CA, USA and CFX Connect Real-time system (Bio-Rad, Hercules, CA, USA) for 45 cycles and fold change for all samples was calculated using the 2^−ΔΔCt^ method. Peptidylprolyl Isomerase A (PPIA) was used as a housekeeping gene for mRNA expression analysis. Primers used for qRT-PCR were described previously [20].

### 2.4. Western Blotting

Western blotting was performed, as previously described [20]. The following primary antibodies were used for western blot analysis: anti-UCP1 (rabbit, Alpha Diagnostic International, San Antonio, TX, USA), cytochrome c oxidase subunit IV (COX IV, rabbit, Cell Signaling, Danvers, MA, USA), Total OXPHOS Rodent WB Antibody Cocktail (mouse, Abcam, Boston, MA, USA), phospho- hormone-sensitive lipase (HSL) (Ser563, rabbit, Cell Signaling, Danvers, MA, USA), HSL (rabbit, Cell Signaling, Danvers, MA, USA), Sirtuin Antibodies (rabbit, Cell Signaling, Danvers, MA, USA), Acetylated Lysine (rabbit, Cell Signaling, Danvers, MA, USA) and α/β tubulin (rabbit, Cell Signaling, Danvers, MA, USA)

### 2.5. Analysis of Mitochondrial Function

Oxygen concentrations and oxygen consumption rates were measured by the Oxygraph plus system (Hansatech, Norfolk, UK) with chart recording software, or Seahorse XF Analyzers [20]. OCRs were normalized according to protein concentrations. Uncoupled respiration was calculated by subtraction of the KCN-induced OCR from the oligomycin A-induced OCR. ATP related respiration was calculated by subtraction of the oligomycin A-induced OCR from the basal OCR.

### 2.6. Statistical Analysis

GraphPad Prism 5 software (GraphPad Software, La Jolla, CA, USA) was used for the statistical analysis. Data are presented as mean ± standard errors of the means (SEMs). Statistical significance between two groups was determined by unpaired *t*-test. Comparisons among multiple groups were performed using a two-way analysis of variance (ANOVA), with Bonferroni post hoc tests to determine *p* values.

## 3. Results

### 3.1. Myricetin Reduced Triglyceride Content in Cultured Adipocytes In Vitro

We first examined the lipid content of adipocytes differentiated from C3H10T1/2 cells after myricetin treatment. Nutrient starvation medium was used as a positive control that stimulates lipid consumption [20]. As shown in Figure 1, myricetin treatment for 24 h reduced the content of neutral lipids by 32.6 ± 0.05%, as indicated by a reduction in the intensity of Boron dipyrromethene fluorophore (BODIPY) staining (Figure 1A,B). The biochemical analysis of TGs further confirmed that myricetin treatment significantly reduced the lipid content in adipocytes by 36.9 ± 0.06%, compared with controls (Figure 1C).

### 3.2. Myricetin Treatment Increases Mitochondrial Content and Mitochondrial Respiration in Adipocytes

PKA-dependent lipolysis is a well-known catabolic pathway that is governed by cytosolic lipases, such as, monoacylclycerol lipase, HSL and adipose triglyceride lipase (ATGL), HSL is a rate-limiting enzyme and the phosphorylation of HSL can be mediated by PKA downstream. Therefore, we examined phosphorylation of HSL. We included the 8Br-cAMP treated condition as positive controls that stimulate PKA-dependent lipolysis [20]. As expected, 8Br-cAMP significantly increased the phosphorylationof HSL compared with the control condition. However, nutrient starvation and myricetin did not affect the levels of HSL activation (Figure 2A). In contrast, mitochondrial content was upregulated by starvation and myricetin treatment, as indicated by immunoblotting for enzymes of the mitochondrial oxidative phosphorylation system (Figure 2B). Myricetin treatment did not affect the expression levels of fatty acid synthase (FAS), an enzyme responsible for de novo lipogenesis, and ATGL (Figure 2C).

To examine the mitochondrial metabolism of adipocytes, we measured the oxygen consumption rate (OCR) of adipocytes treated with myricetin or vehicle (Figure 3A,B). As shown in Figure 3, both the basal mitochondrial respiration rate and the ATP-linked respiration rate were increased by myricetin treatment. In addition to PKA-dependent lipolysis, UCP1-dependent, non-shivering thermogenesisin brown adipocytes accounts for lipid consumption. As brown adipocytes can be characterized by UCP1 expression and an increase in the content of mitochondria, we examined the expression of brown adipocyte markers. The expression levels of UCP1 and other brown adipocyte markers such as deiodinase 2 (Dio2), and elongation of very long chain fatty acids protein 3 (Elovl3)) were not upregulated (Figure 3C). The expression levels of mitochondrial marker, Cox8b, and a transcription factor that induces mitochondrial biogenesis, Ppargc1a, were slightly upregulated in myricetin treated groups (Figure 3C). In addition, mitochondrial labeling was performed by using MitoTracker, which is a mitochondrion-specific fluorescence probe and mitochondrial membrane potential indicator. As shown in Figure 3D, myricetin treatment increased mitochondrial membrane potential in differentiated adipocytes from C3H10T1/2 cells compared to control conditions. Thus, these data suggest that the upregulation of brown adipocyte marker expression did not appear to contribute to the loss of lipids, rather, the increase in mitochondrial content and metabolic activity was mediated by the independent expression of thermogenic genes.

### 3.3. Myricetin Increased Sirt1, Sirt3, and Sirt5 Expression in Adipocytes

The sirtuin family has been reported as a regulator of mitochondrial function and mitochondrial biogenesis. As myricetin affects mitochondrial content and mitochondrial OCR in adipocytes, we examined the expression of sirtuin proteins in adipocytes as potential indicators of the downstream mechanisms of the effects of myricetin. As shown in Figure 4, western blotting analysis indicated that Sirt6 and Sirt7 expressions were not affected; however, slight upregulation of Sirt1 was observed. Furthermore, the expression levels of mitochondrial sirtuins, Sirt3 and Sirt5, were significantly upregulated by myricetin treatment (Figure 4A–C). These data suggested that the induction of mitochondrial sirtuins by myricetin was related to the increase in mitochondrial respiration.

### 3.4. Myricetin Increase Mitochondrial Metabolism in Adipocytes through an Increase in Sirt3 Activity

For further examination of the effects of myricetin in a mouse model on obesity, we used a HFD-fed mouse model (Figure 5A). There was no difference in food consumption among the groups. After 2 weeks of myricetin treatment, mice body weight was significantly reduced (Figure 5B) and diet-induced insulin resistance was corrected (Figure 5C). Also, myricetin treatment reduced respiratory exchange rate (RER) in HFD mice (Figure 5D), indicating that myricetnin treatment increased fatty acid consumption. These data suggested that myricetin treatment increased mitochondrial fatty acid oxidation in vivo. In addition, serum TG and FFA levels were reduced by myricetin treatment compared to vehicle controls. Consistent with the in vitro data, Sirt3 and Sirt5 expression were upregulated in mice treated with myricetin (Figure 6A). The effect on Sirt3 expression was more prominent in vivo. The deacetylation of mitochondrial enzymes by Sirt3 has been identified as a mechanism through which mitochondrial oxidative respiration is increased [14]. To determine the deacetylation activity of Sirt3 on mitochondrial proteins, we examined the level of acetylated lysine of mitochondrial fraction of adipose tissue. The acetylated form was reduced by myricetin treatment (Figure 6B). Finally, the siRNA knockdown of Sirt3 expression in fully differentiated adipocytes from C3H10T1/2 cells reduced mitochondrial oxygen consumption rates (Figure 6C,D). These data indicated that the induction of Sirt3 by myricetin contributed to the upregulation of mitochondrial lipid catabolism and exerted anti-obesity effects.

## 4. Discussion

Myricetin, a naturally occurring polyphenol, effectively confers a variety of health benefits not limited to its free radical scavenging antioxidant effect [12]. Beneficial effects of myricetin in rodent model of metabolic disease have been demonstrated by previous works [6,11]. For example, myricetin treatment was reported to ameliorate hyperglycemia in diabetic rats [6]. It displayed a positive effect in controlling weight gain and improved insulin resistance, glucose tolerance, and, hepatic steatosis in db/db mice [6]. In the same model, myricetin stimulated brown adipose tissue and triggered the formation of beige adipose tissue, resulting in increased thermogenic protein expression and mitochondrial biogenesis in inguinal adipose tissue. However, the effect of myricetin on the lipid metabolism of adipocytes remained unknown.

The current study demonstrated that myricetin reduced the lipid content of adipocytes and exerted anti-obesity effects through the upregulation of Sirt3 expression. Mechanistically, myricetin treatment reduced lipid content by increasing mitochondrial oxidative metabolism but did not affect the protein expression levels of major lipolysis enzymes including p-HSL/HSL and ATGL. However, we cannot rule out the possibility that myricetin treatment affects metabolic activity of the major lipases (i.e., ATGL, HSL). Further analyses of the effects of myricetin on the lipase activity are required to determine whether myricetin treatment alter lipolytic response of adipocytes to provide substrates for mitochondrial oxidation. Also, pharmacological lipase inhibitors would be useful to determine the effects of myricetin on TG lipase activity in adipocytes.

Our in vivo data demonstrated that myricetin treatment improved glucose tolerance and increased fatty acid consumption of HFD-fed mice determined by indirect calorimetry. Moreover, myricetin treatment reduced serum TG and FFA levels of HFD- fed mice. Although these are important pieces of information that suggested potentially beneficial effects of myricetin on blood lipid profiles, this study did not directly address the roles of myricetin in regulating the lipolysis rate of adipose tissue. Apart from its effects on lipid catabolism in adipocytes, it is likely that myricetin may alter intestinal absorption processes, and FFA uptake into adipocytes. Additional studies that directly measure lipid flux in vivo [21], such as metabolic kinetic study using tracer, will be required to fully understand the mechanisms and beneficial effects of myricetin.

This study focused on the role of myricetin in mitochondrial oxidative metabolism in relation to Sirt3 activity; however, lipid content can be altered by regulation of *de novo* lipogenesis and FFA uptake. Although myricetin treatment did not affect fatty acid synthase (FAS) expression, further investigation on the role of myricetin in *de novo* lipogenesis is required to fully understand molecular mechanisms by which myricetin reduces lipid content in adipocytes. For example, the analysis of acyl CoA carboxylase activity [22] would determine its effects on *de novo* lipogenesis. Likewise, lipoprotein lipase is required for the uptake of fatty acids into adipocytes by vascular TG hydrolysis [21]. In future studies, it would be informative to determine whether myricetin treatment affects the activity of lipoprotein lipase (LPL).

The central finding of this study is the upregulation of Sirt3 expression as a molecular mechanism of the anti-obesity effects of myricetin. Sirtuins are NAD+-dependent protein-modifying enzymes with deacetylating, desuccinylating, and demalonylating effects, and are proficient in the enhancement of mitochondrial biogenesis, lipid metabolism, and insulin secretion and sensitivity [13,23]. These enzymes are sensors of cellular energy balance, which regulate the changes to the metabolic responses triggered by nutritional and stress factors in tissues [24]. SIRT3, a key player in the deacetylation of mitochondrial proteins, increases cellular ATP generation and mitochondrial contentsduces mitochondrial biogenesis [25]. The downregulation of SIRT3 results in the hyperacetylation of mitochondrial proteins, leading to a rise in metabolic defects triggered by the stunted function of mitochondria [14,26]. As a deacetylase, activation of Sirt3 results in a positive impact on mitochondrial homeostasis, including mitochondrial oxidative phosphorylation [14]. As a regulator of mitochondrial metabolism, SIRT3 activates acetyl-CoA synthetase 2 (AceCS2)by deacetylation [27] and regulates ATP synthesis in mitochondria through the deacetylation of proteins in the electron transport chain complexes, 3-hydroxy-3-methylglutaryl CoA synthase 2 (HMGCS2), and long-chain acyl-CoA dehydrogenase (LCAD) [28]. Further study is required to determine other substrates of Sirt3 in adipocytes that were induced by myricetin treatment. Although we focused on the mitochondrial function of SIRT3 in this study, myricetin increased SIRT1 and SIRT5 expression levels. Thus, the histone and non-histone targets of other sirtuins induced by myricetin treatment could be further investigated, which may enable the discovery of the novel regulatory pathways of lipid catabolism mediated by myricetin treatment and other compounds that mimic calorie restriction.

Collectively, the current study demonstrates that the anti-obesity effect of myricetin was mediated through the upregulation of Sirt3 expression and the subsequent activation of mitochondrial fatty acid oxidation. The effect of specific natural compounds on adipocytes deserves further investigation as a therapeutic target for the prevention and treatment of obesity-related metabolic disease.

## Figures and Tables

**Figure 1 nutrients-10-01962-f001:**
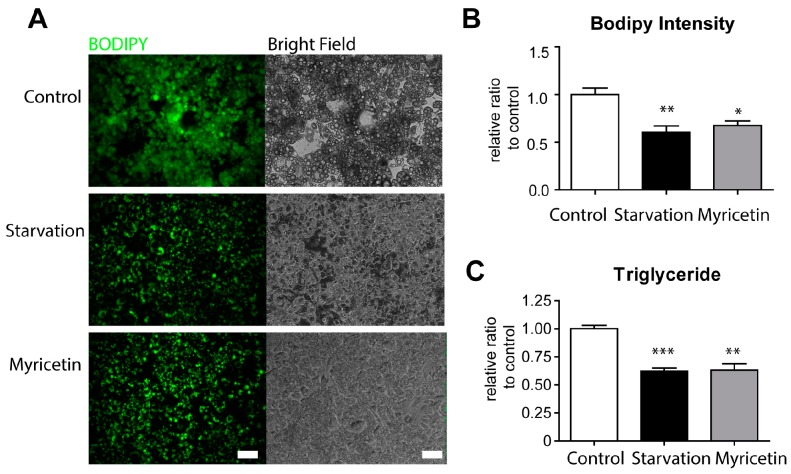
Effects of myricetin on lipid content in adipocytes differentiated from C3H10T1/2 cells. (**A**) Boron dipyrromethene fluorophore (BODIPY) staining in adipocytes differentiated from C3H10T1/2 cells treated with myricetin (10 μM) for 24 h. Nutrient starvation medium (starvation) was used as positive controls. Size bar = 40 μm (**B**) Quantification of BODIPY intensity of (**A**). (**C**) Intracellular triglyceride (TG) levels. (*n* = 3, means ± SEM, * *p* < 0.05, ** *p* < 0.01, *** *p* < 0.001)

**Figure 2 nutrients-10-01962-f002:**
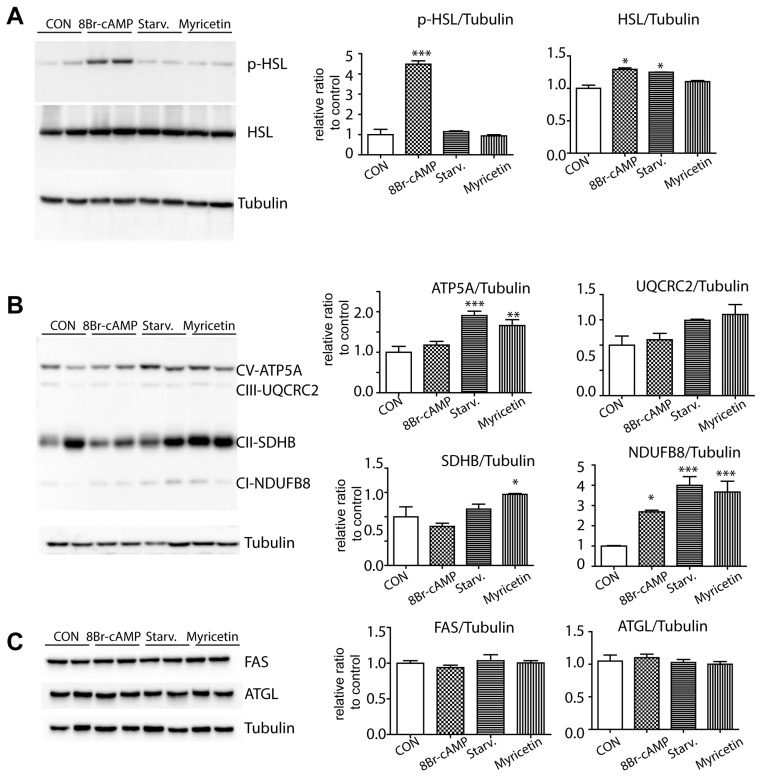
In vitro effects of myricetin on protein kinase A (PKA) signaling and mitochondrial content in adipocytes. (**A**–**C**) Immunoblot analysis of phosphor form of hormone sensitive lipase (p- hormone-sensitive lipase (HSL)), HSL, mitochondrial enzymes involved in oxidative phosphorylation, fatty acid synthase (FAS), and adipose triglyceride lipase (ATGL) in adipocytes differentiated from C3H10T1/2 cells treated with vehicle controls (CON), starvation medium (Starv.), 8Br-cAMP (1 mM) and myricetin (10 μM) for 24 h. (*n* = 4, means ± SEM, * *p* < 0.05, ** *p* < 0.01, *** *p* < 0.001).

**Figure 3 nutrients-10-01962-f003:**
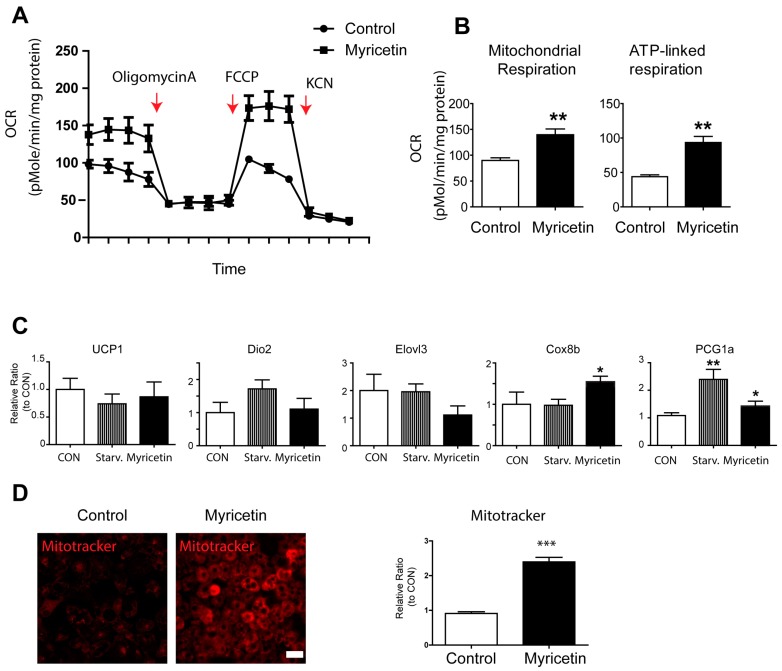
In vitro effects of myricetin on mitochondrial respiration and brown adipocyte specific marker expression in adipocytes. (**A**) Analysis of oxygen consumption rate (OCR) of adipocytes differentiated from C3H10T1/2 cells treated with vehicle controls (CON), and myricetin (10 μM) for 24 h with a series of treatments of indicated drugs (Oligomycin, Carbonyl cyanide-4-(trifluoromethoxy)phenylhydrazone (FCCP), and potassium cyanide (KCN)). Arrows indicate the time of the treatments. (**B**) Comparisons of basal mitochondrial respiration and ATP-linked respiration between cells treated with myricetin and vehicle controls. (two-tailed unpaired *t*-test). (**C**) Quantitative PCR analysis of the genes involved in mitochondrial respiration and brown adipocyte specific markers in adipocytes differentiated from C3H10T1/2 and treated with myricetin (10 μM) for 24 h (**D**) MitoTracker staining of adipocytes differentiated from C3H10T1/2 cells treated with myricetin (10 μM) for 24 h and vehicle controls (*n* = 3, means ± SEM, * *p* < 0.05, ** *p* < 0.01, *** *p* < 0.001). Size bar = 20 μm.

**Figure 4 nutrients-10-01962-f004:**
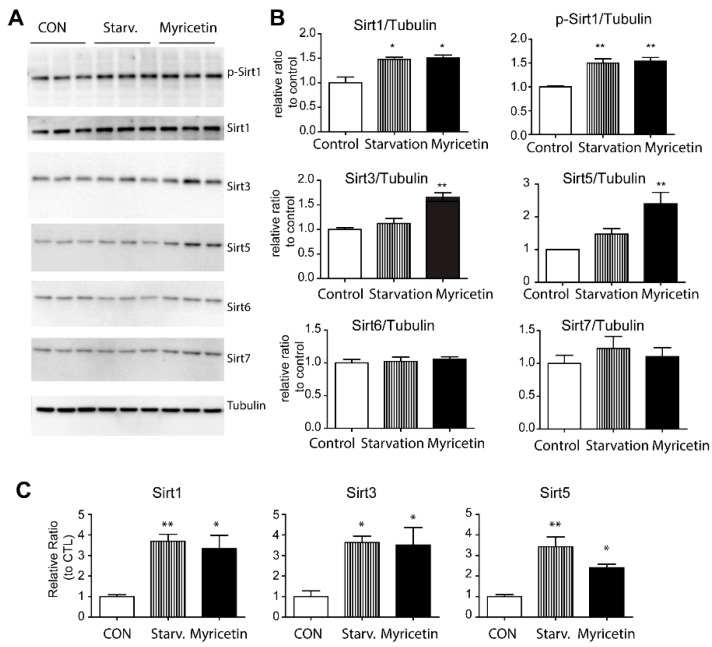
Effect of myricetin on mitochondrial sirtuin expression in cultured adipocytes. (**A**,**B**) Immunoblot analysis and quantification of sirtuins in adipocytes differentiated from C3H10T1/2 cells treated with vehicle controls (CON), starvation medium and myricetin (10 μM) for 24 h. (**C**) qPCR analysis of sirtuins in adipocytes differentiated from C3H10T1/2 treated with myricetin (10 μM) for 24 h (*n* = 3, means ± SEM, * *p* < 0.05, ** *p* < 0.01).

**Figure 5 nutrients-10-01962-f005:**
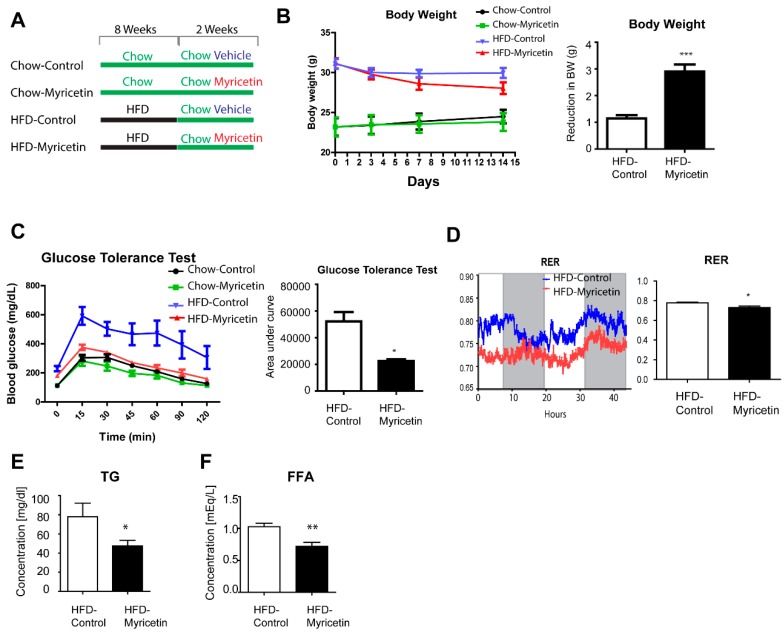
Effect of myricetin on body weight and glucose tolerance of mice fed with a high fat diet. (**A**) Study design for 8 weeks of high fat diet feeding and 2 weeks of myricetin treatment. (**B**–**D**) Body weight (**B**), glucose tolerance test (**C**), and respiratory exchange rate (RER): dark, 12 h; light, 12 h (**D**) of control high fat diet (HFD)-fed mice and HFD-fed mice treated with myricetin for 2 weeks. Analysis of triglyceride (TG) (**E**) and free fatty acid (FFA) (**F**) levels in serum obtained from the HFD-fed mice and HFD-fed mice treated with myricetin for 2 weeks (*n* = 4, means ± SEM, * *p* < 0.05, ** *p* < 0.01, *** *p* < 0.001).

**Figure 6 nutrients-10-01962-f006:**
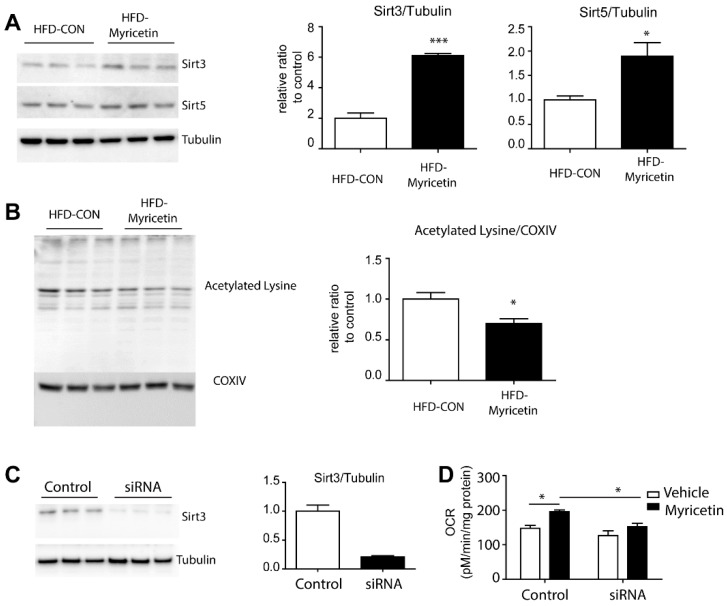
Effect of myricetin on Sirt3 expression and lysine acetylation level in inguinal adipose tissue of HFD-fed mice. (**A**) Immunoblot analysis of Sirt3, and Sirt5 in white inguinal adipose tissue of mice fed with HFD for 8 weeks. (**B**) Immunoblot analysis of acetylated lysine in mitochondrial fraction of white inguinal adipose tissue of mice fed with HFD for 8 weeks. (**C**) Immunoblot analysis of Sirt3 expression in adipocyte culture treated with siRNA targeting Sirt3. (**D**) Analysis of myricetin effect on OCR of fully differentiated adipocytesfrom C3H10T1/2 cells treated with siRNA targeting Sirt3. Myricetin was treated at the concentration of 10 μM for 24 h. (*n* = 3, means ± SEM, * *p* < 0.05, *** *p* < 0.001)).

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
