# Peer review of "Myricetin Exerts Anti-Obesity Effects through Upregulation of SIRT3 in Adipose Tissue"

_nutrients, 2018, doi:10.3390/nu10121962_

Reviewer 1 Report

In this manuscript, Akindehin et al. investigate the molecular mechanism behind the activity exerted by Myricetin on fatty acid metabolism.

1) The authors start using adipocytes differentiated from C3H10T1/2 cells. Myricetin decreases total TGs in these cells. The decreased amount of TGs is not the consequence of an increased lipase activity since phosphorylation of HSL is not altered by myricetin. In contrast, mitochondrial OXPHOS proteins are upregulated after 24 hours of treatment with myricetin. Similar is the effect of myricetin on overall mitochondrial activity and mitochondrial biogenesis. The authors conclude that myricetin increases mitochondrial content.

2) In the second part of the manuscript, the authors employed a high-fat diet (HFD)-fed mouse model to determine the anti-obesity effects of myricetin. After 2 weeks of myricetin treatment, they noticed that body weight was significantly reduced and diet-induced insulin resistance was corrected. Finally, the authors move to investigate the role of Sirtuins in mediating myricetin effect. 24 hours of treatment with myricetin are sufficient to upregulate Sirt3 and Sirt5. By silencing Sirt3, they conclude that at least this Sirtuin is involved in the cascade of metabolic reaction induced by myricetin.

The effect of Myricetin on fatty acid metabolism is not novel in the field (see for example http://dx.doi.org/10.1155/2012/787152). However the manuscript presents new insights, is interesting and technically sound. Upon a major revision, this reviewer thinks that the manuscript can be accepted for publication in Nutrients.

Major points:

Major point 1: The increased expression of Cox8b is not sufficient to prove mitochondrial biogenesis. The authors should investigate if myricetin increases the number of mitochondria by using a mitochondrial die, like they did to stain neutral lipids in Figure 1. (an option would be one of the mitotrackers commercially available ). 

Major point 2: Myricetin could have a direct effect on mitochondrial activity (acting like uncoupler for example). (See Kopustinskiene et al. http://dx.doi.org/10.1155/2015/232836 ). Considering the short incubation time necessary for myricetin to act, (24 hours of incubation, a time barely sufficient for inducing protein synthesis or mitochondrial biogenesis ) the author should measure the effect of shorter incubation with myricetin. Is a 6 or 12 hours incubation time sufficient to boost mitochondrial activity? Would the Seahorse assay change upon short incubation times with myricetin? If Myricetin acts even after short incubation time,  Sirt3 activation should than be considered a consequence of myricetin activity on mitochondria and not the mediator of its effect.

Major point 3: How does the Seahorse assay look like in silenced Sirt3 cells? The remaining activity of myricetin in siSirt3 cells is due to Sirt5 induction by Myricetin?

Minor points:

line 34  References in the introduction must be definitely updated with new reviews on obesity.

line 44  The abbreviation UCP1 is not used everywhere in the text. At line 187 the abbreviation is not used

line 76  The manuscript would benefit of a clear indication of the composition of the HFD and of few more references on the effect of this diet on the weight of mice

line 204  Sirt1 is incorrectly written

Reference 5 and 18 seem to overlap. Which one is correct? 

Author Response

Major points:

Major point 1: The increased expression of Cox8b is not sufficient to prove mitochondrial biogenesis. The authors should investigate if myricetin increases the number of mitochondria by using a mitochondrial die, like they did to stain neutral lipids in Figure 1. (an option would be one of the mitotrackers commercially available ). 

Response 1: As suggested by the reviewer, MitoTracker fluorescence images are included in this revision (Figure 3D). The data indicated that myricetin increased mitochondrial membrane potentials in adipocytes differentiated from C3H10T1/2 cells. In our original manuscript, the enzymes involved in mitochondrial oxidative phosphorylation (OXPHOS enzymes) were also measured (Figure 2B).

Figure 3 (D) Mitotracker staining of cells treated with myricetin and vehicle controls (10mM) for 24 h.

Major point 2: Myricetin could have a direct effect on mitochondrial activity (acting like uncoupler for example). (See Kopustinskiene et al. http://dx.doi.org/10.1155/2015/232836 ). Considering the short incubation time necessary for myricetin to act, (24 hours of incubation, a time barely sufficient for inducing protein synthesis or mitochondrial biogenesis ) the author should measure the effect of shorter incubation with myricetin. Is a 6 or 12 hours incubation time sufficient to boost mitochondrial activity? Would the Seahorse assay change upon short incubation times with myricetin? If Myricetin acts even after short incubation time,  Sirt3 activation should than be considered a consequence of myricetin activity on mitochondria and not the mediator of its effect.

Response 2: We really appreciated the reviewer’s constructive comments. To test the direct effect of myricetin activity on mitochondria as suggested by the reviewer, we performed OCR measurement by Seahorse assay with cells treated with myricetin for 6 hours. However, we found no statistically significant difference between control and treated groups as shown below.

Although we agree that 24hr might not be enough for mitochondrial biogenesis, mitochondrial contents can be regulated by multiple modes, such as regulation of mitochondrial fission/fusion and mitochondrial degradation rate (organelle and protein turn over, mitophagy) (ref.: Dominy, J. E. and P. Puigserver (2013). "Mitochondrial Biogenesis through Activation of Nuclear Signaling Proteins." Cold Spring Harbor Perspectives in Biology 5(7)). Moreover, the time for new protein synthesis can vary depending on the type stimuli. For example, it has been reported that increase in UCP1 protein levels can be detected within an hour (ref.: Santos, R. S., et al. (2018). "Activation of estrogen receptor alpha induces beiging of adipocytes." Molecular Metabolism 18: 51-59.)

Major point 3: How does the Seahorse assay look like in silenced Sirt3 cells? The remaining activity of myricetin in siSirt3 cells is due to Sirt5 induction by Myricetin?

Response 3: As the reviewer suggested, we performed OCR measurement to test the effects of myricetin on oxygen consumption of Sirt3-silenced cells (Supplemental Figure S1). The basal/maximal levels of OCR and ATP production by mitochondrial respiration were reduced in siRNA-treated groups, supporting that upregulation of mitochondrial activity by myricetin treatment requires Sir3 expression. As the reviewer pointed out, the remaining activity of myricetin is possibly due to Sirt5 expression. Also, it is likely that remaining Sirt3 (~20% of control Sirt3 expression due to technical limitation of siRNA silencing) might be responsible for myricetin activity.

Minor points:

line 34  References in the introduction must be definitely updated with new reviews on obesity.

Response: We updated the introduction with a new recent review paper on obesity {Heymsfield, S.B.; Wadden, T.A. Mechanisms, pathophysiology, and management of obesity. The New England journal of medicine 2017, 376, 254-266.}

line 44  The abbreviation UCP1 is not used everywhere in the text. At line 187 the abbreviation is not used

Response: We thank the reviewer for the correction. We corrected to use the abbreviation (UCP1) at line 187.

line 76  The manuscript would benefit of a clear indication of the composition of the HFD and of few more references on the effect of this diet on the weight of mice

Response: A suggested by the reviewer, the composition of HFD is indicated at line 76. New references that reported the effect of this diet are also included in this revised manuscript.

line 204  Sirt1 is incorrectly written

Response: We thank the reviewer for the correction. We corrected the typographical error (Sir1 -> Sirt1).

Reference 5 and 18 seem to overlap. Which one is correct? 

Response: We thank the reviewer for the correction. We removed the overlapping reference in this revised manuscript.

Reviewer 2 Report

The manuscript is well designed and having organized presentation of data and scientific soundness. The introduction clearly establishes the theoretical background to the work. The materials and methods are appropriate and the science is of an appropriately high standard for a preliminary study.  The results are presented clearly and the excellent figures put them clearly into content.  The findings are well discussed and the results support the conclusions. The only concern with the study about the blood brain barrier penetration and bioavailability of myricetin related with the anti-obesity effects. There are few typo errors in the sentences which are needed to corrects. There are few minor issues requires to be corrected.

Page 1; Line 23: Mention the dose of myricetin.

Page 1; Line 24: How much significant in percentage?

Page 1; Line 28: Describe the results in numerical data in brief

Page 1; Line 34: The reference style is incorrect and should be before full stop throughout the manuscript.

Page 2; Line 59: Need coma after reference.

Page 2; Line 91: How much cells (number) were used in assay.

Page 4; Line 153: Instead of graphical presentation, the results should describe in brief sentences.

Author Response

Page 1; Line 23: Mention the dose of myricetin

Response: We thank the reviewer for comments. We included the dose of myricetin in the abstract.

Page 1; Line 24: How much significant in percentage?

Response: We thank the reviewer for comments. We included the percentage in the abstract. (“The in vivo treatment reduced body weight by 11%”)

“ Page 1; Line 28: Describe the results in numerical data in brief

Response: We thank the reviewer for comments. We included the numerical data in the abstract. (“…. reduced…. by 27% ,compared to controls)

Page 1; Line 34: The reference style is incorrect and should be before full stop throughout the manuscript.

Response: We thank the reviewer for the correction. We corrected the reference style throughout the manuscript

Page 2; Line 59: Need coma after reference.

Response: We thank the reviewer for the correction. We corrected the reference style throughout the manuscript

Page 2; line 91: How much cells (number) were used in assay.

Response: As the reviewer suggested, we included the number of the cells used in the assay.

Page 4; Line 153: Instead of graphical presentation, the results should describe in brief sentences.

Response: As the reviewer suggested, we describe the results: “ myricetin treatment for 24 h reduced the content of neutral lipids by 32%.” and “myricetin treatment significantly reduced the lipid content in adipocytes by 36% compared to controls.”

Reviewer 3 Report

The authors examined the anti-obesity effect of myricetin in vitro and in vivo. The study is well done, however not complete.   The authors provide too less evidence to establish a link between myricetin-induced fat mass reduction and incraesed mitochondrial activity. More specifically, reduced triglyceride content of cells and decreased body weight are not sufficient. Mechanisms should be addressed as indicated below.

Major issues:

Fig. 1 shows decreased triglycerides (TAGs) in myricetine treated cells but unchanged p-HSL. This is discrepant. For clarification the authors should measure TAG activity to provide ecvidence for enhanced lipolysis being responsible for the decreased TAG. Additionally, glucose uptake should be examined to assess whether decreased uptake contributes to lower TAGs.

Fig. 5:

-Please indicate whether food concuptio was comparable in all treatment groups during 10 weeks of treatment.

-in addition to body weight the authors should provide data on fed and fasted lipid levels including plasma TAG and FFA.

-the authors should analyse p-HSL/HSL, ATGL and lipoprotein lipase (LPL)  content of fat tissue, TAG activity in fat tissue of the treated mice. TAG assay should be done in the presence or absence of appropriate inhibitors of HSL, ATGL or LPL. Only by such an approach the authors can completely clarify the anti-obesity effect  of myricetin.

Minors:

1. abbreviations should be defined in the text and fig legends of Fig 2 and 3.

2. Discussion is to general and not sufficiently related to the results.

Author Response

Major issues:

Fig. 1 shows decreased triglycerides (TAGs) in myricetine treated cells but unchanged p-HSL. This is discrepant. For clarification the authors should measure TAG activity to provide ecvidence for enhanced lipolysis being responsible for the decreased TAG. Additionally, glucose uptake should be examined to assess whether decreased uptake contributes to lower TAGs.

Response: We thank the reviewer for the constructive comments. As suggested by the reviewer, glucose uptake was measured by incubating with 100mM 2-(N-(7-Nitrobenz-2-oxa-1,3-diazol-4-yl)Amino)-2-Deoxyglucose (2-NBDG) for 1 hr (ref: Zou, C.; Wang, Y.; Shen, Z. 2-nbdg as a fluorescent indicator for direct glucose uptake measurement. Journal of biochemical and biophysical methods 2005, 64, 207-215.) and we found no difference in NBDG fluorescence levels between myricetin-treated groups and controls.

Also, we measured protein levels of fatty acid synthase (FAS), an enzyme responsible for de novo lipogenesis, by immunoblotting and found no significant differences between grorups (Figuer 2C). As the reviewer pointed out, it is informative for interpretation of the results; thus, we included the data in this revision

Fig. 5:

-Please indicate whether food concuptio was comparable in all treatment groups during 10 weeks of treatment.

Response: We thank the reviewer for the comments. We indicated that there was no difference in food consumption among groups (at line 233 in the revised manuscript).

-in addition to body weight the authors should provide data on fed and fasted lipid levels including plasma TAG and FFA

Response: As the reviewer pointed out, information of blood lipid is important. Therefore, we included data of serum lipid levels at fed state (TAG and FFA), indicating that myricetin reduced plasma TAG and FFA levels.

-the authors should analyse p-HSL/HSL, ATGL and lipoprotein lipase (LPL)  content of fat tissue, TAG activity in fat tissue of the treated mice. TAG assay should be done in the presence or absence of appropriate inhibitors of HSL, ATGL or LPL. Only by such an approach the authors can completely clarify the anti-obesity effect  of myricetin.

Response: As suggested by reviewer, we measured p-HSL/HSL, and ATG, and we did not observe significant increase in those two major lipases. Thus, we proposed that myricetin increase mitochondrial beta oxidation of FFA and decreased TG content in adipose tissue without affecting canonical lipolysis pathway.

Also, we agreed that pharmacological inhibitor experiments would be informative if myricetin works through specific lipases. However, as we did not observe increase in lipase expression levels, we did not further investigate the effects of the pharmacological inhibitors of HSL or ATGL.

The reviewer is right that the major lipases pHSL/HSL and ATGL are important to increase lipolysis; however, the increase in mitochondrial beta-oxidation would be another pathway that can reduce FFA levels and lipid content. Also, it is important to note that pharmacological inhibition of ATGL, not the activation of ATGL, has been investigated as a novel candidate drug that increases insulin sensitivity (Schweiger, M., (2017). Pharmacological inhibition of adipose triglyceride lipase corrects high-fat diet-induced insulin resistance and hepatosteatosis in mice. Nature Communications 8, 14859). It is because overproduction of FFA by excessive lipolysis could be harmful, causing lipotoxicity (Ref: 10.1016/jNaNet.2011.12.018: Zechner, R.,  et al (2012). FAT SIGNALS--lipases and lipolysis in lipid metabolism and signaling. Cell metabolism 15, 279-291.). Also, FFA generated by activation of PKA-HSL activity can be re-esterified into TG. Although the lipase activity is important for the reduction of lipid content, mitochondrial beta-oxidation is also one of the important catabolic branches of the lipid metabolism. Our results indicated that myricetin mainly affect mitochondrial FFA metabolism.

We also appreciated the reviewer’s concerns on the potential role of myricetin in LPL activity. We agree that LPL activity is important in adipose tissue metabolism, but this study focused on the effect of myricetin on mitochondrial function of adipocytes. Thus, we indicated the limitation of this study in the discussion and mentioned that future study is needed to determine the effect of myricetin on LPL (at line 280-293). Additionally, although it is well accepted that LPL is required for the uptake of fatty acids into adipocytes by vascular TG hydrolysis, it might not directly affect adipose tissue mass (Weinstock, P.H et.al., (1997). Lipoprotein lipase controls fatty acid entry into adipose tissue, but fat mass is preserved by endogenous synthesis in mice deficient in adipose tissue lipoprotein lipase. PNAS 94, 10261-10266). 

Minors:

abbreviations should be defined in the text and fig legends of Fig 2 and 3.

Response: As the reviewer suggested, abbreviations are defined in this revision: i.e. HSL (hormone-sensitive lipase), adipose triglyceride lipase (ATGL)  and Carbonyl cyanide-4-(trifluoromethoxy) phenylhydrazone (FCCP),

2. Discussion is to general and not sufficiently related to the results.

Response: As the reviewer suggested, we modified the discussion including specific interpretation of the data. (line 285~297)

. “Mechanistically, myricetin treatment reduced lipid content by increasing mitochondrial oxidative metabolism, but did not affect the major lipolysis enzymes including p-HSL/HSL and ATGL. It was consistent with the in vivo data demonstrating that myricetin treatment reduced serum TG and FFA levels. Because the excessive lipolysis can cause lipotoxicity[21], reduction in lipid content without increasing lipolysis-mediated FFA efflux could be a beneficial feature of myricetin as a potential anti-obesity therapeutics. This study focused on the role of myricetin in mitochondrial oxidative metabolism in relation to Sirt3 activity; however, lipid content can be altered by regulation of de novo lipogenesis and FFA uptake. Although myricetin treatment did not affect FAS expression, further investigation on the roles of myricetin in de novo lipogenesis is required to fully understand molecular mechanisms by which myricetin reduces lipid content in adipocytes. The effect of myricetin on lipoprotein lipase (LPL) activity has not been studied in the current work. As LPL is required for the uptake of fatty acids into adipocytes by vascular TG hydrolysis[21], in future studies, it would be informative to determine whether myricetin treatment affects the expression and activity of LPL in adipose tissue.”

Round  2

Reviewer 1 Report

The authors have fulfilled all my requests and have replied to all my criticisms.

Author Response

We thank the reviewer for insightful suggestions and constructive comments

Reviewer 3 Report

My major concern regarding this paper remains, despite additional experiments which are not correctly done or interpreted. Accordingly there is still completely not clear  which fatty acids are consumed by activated mitochondria, those increasingly taken up by adipose tissue or those generated by enhanced lipolysis. Therefore, it is still not clear what is exact anti-obesity mechanism of myricetin. On one hand  myricetin induces oxidative phosphorylation in mitochondria, but it is not clear what is a source of fuel used by activated mitochondria.

The authors say in the abstract that the aim was to study the effect of m. on lipid metabolism. However, by showing decreased TG in cultured cells and Western blots of enzymes like ATGL, HSL or FAS (Fas is  not rate limiting enzymes, rather acyl CoA carboxylase), lipid metabolism is not properly addressed.

lanes 370-375: this part of discussion is not correct because: 1. the levels of enzymes ecxamined by Western blot say northing about activity of those enzymes and 2. decreased plasma TG and FFA levels in FED state say nothing about adipose lipolysis rate. Decreased FFA and TG may reflect altered intestinal absorption, increased LPL activity or increased uptake into cells.

If the authors can not perform suggested experiments (TG lipase activity in cells and adipose tissue, LPL activity in post-heparin plasma, plasma FFA and TG in fasted state) than

lipid metabolism should not be highlighted as the aim of the study.

In a very clear limitation section of the discussion the authors should provide a clear explanation what can not be concluded regarding lipid metabolism and which experiments should be done to obtain a clear picture on the anti-obesity effect of m.

Author Response

Point-by-point responses

We thank the reviewer for the insightful suggestions and the opportunity to improve our manuscript

My major concern regarding this paper remains, despite additional experiments which are not correctly done or interpreted. Accordingly there is still completely not clear  which fatty acids are consumed by activated mitochondria, those increasingly taken up by adipose tissue or those generated by enhanced lipolysis. Therefore, it is still not clear what is exact anti-obesity mechanism of myricetin. On one hand  myricetin induces oxidative phosphorylation in mitochondria, but it is not clear what is a source of fuel used by activated mitochondria.

Response: We agree with the reviewer’s opinion pointing out that our current study mainly focused on mitochondrial function, but did not clearly address the source of FFAs that are utilized by mitochondrial oxidation in response to myricetin treatment. Therefore, we incorporated the reviewer’s concerns in the manuscript and discussed about the limitation of our study in the revised manuscript (lines 298-303) .

Line 298-303: However, we cannot rule out the possibility that myricetin treatment affects metabolic activity of the major lipases (i.e. ATGL, HSL). Further analyses of the effects of myricetin on the lipase activity are required to determine whether myricetin treatment alter lipolytic response of adipocytes to provide substrates for mitochondrial oxidation. Also, pharmacological lipase inhibitors would be useful to determine the effects of myricetin on TG lipase activity in adipocytes.

The authors say in the abstract that the aim was to study the effect of m. on lipid metabolism. However, by showing decreased TG in cultured cells and Western blots of enzymes like ATGL, HSL or FAS (Fas is  not rate limiting enzymes, rather acyl CoA carboxylase), lipid metabolism is not properly addressed.

Response: As we agree with the reviewer’s comments, we modified the abstract, indicating that “ this study aimed to examine the effects of myricetin on the expression levels of genes involved in lipolysis…” ,but not “lipid metabolism”

Line 20-21: “This study aimed to examine the effects of myricetin on expression levels of genes involved in lipolysis and mitochondrial metabolism in adipocytes and the anti-obesity potential of myricetin.”

lanes 370-375: this part of discussion is not correct because: 1. the levels of enzymes ecxamined by Western blot say northing about activity of those enzymes and 2. decreased plasma TG and FFA levels in FED state say nothing about adipose lipolysis rate. Decreased FFA and TG may reflect altered intestinal absorption, increased LPL activity or increased uptake into cells.

As the reviewer suggested, we corrected our interpretation.

Lines 306-311: “Moreover, myricetin treatment reduced serum TG and FFA levels of HFD- fed mice. Although these are important pieces of information that suggested beneficial effects of myricetin on blood lipid profiles, this study did not directly address the roles of myricetin in regulating lipolysis rate of adipose tissue. Apart from its potential effects on lipid catabolism in adipocytes, it is likely that myricetin may alter intestinal absorption processes, and FFA uptake into adipocytes. Additional studies that directly measure lipid flux in vivo [ref. : Previs, S.F., et al. New methodologies for studying lipid synthesis and turnover: Looking backwards to enable moving forwards. Biochimica et Biophysica Acta (BBA) - Molecular Basis of Disease 2014, 1842, 402-413], such as metabolic kinetic study using tracer, will be required to fully understand mechanisms of beneficial effects of myricetin.”

If the authors can not perform suggested experiments (TG lipase activity in cells and adipose tissue, LPL activity in post-heparin plasma, plasma FFA and TG in fasted state) than

lipid metabolism should not be highlighted as the aim of the study.

As the reviewer suggested, we have modified the abstract, and discussion to clarify our interpretation. Specifically, we corrected the “lipid metabolism” into “the expression levels of genes involved in lipolysis” to avoid misinterpretations

In a very clear limitation section of the discussion the authors should provide a clear explanation what can not be concluded regarding lipid metabolism and which experiments should be done to obtain a clear picture on the anti-obesity effect of m.

As the reviewer suggested, we indicated examples of experiments that should be done to understand mechanisms of the potential anti-obesity effect of myricetin

Lines 300-303: Further analyses of the effects of myricetin on the lipase activity are required to determine whether myricetin treatment alter lipolytic response of adipocytes to provide substrates for mitochondrial oxidation. Also, pharmacological lipase inhibitors would be useful to determine the effects of myricetin on TG lipase activity in adipocytes.

Line .310:  Additional studies that directly measure lipid flux in vivo, such as metabolic kinetic study using tracer, will be required to fully understand mechanisms of beneficial effects of myricetin.

Line 319: ….analysis of acyl CoA carboxylase activity would be informative to determine its effects on de novo lipogenesis.

Line 322: in future studies, it would be informative to determine whether myricetin treatment affects LPL activity.

Round  3

Reviewer 3 Report

The authors have implemented all suggestions and improved the manuscript significantly.